# January 2016 extensive summer melt in West Antarctica favoured by strong El Niño

Julien P. Nicolas[1], Andrew M. Vogelmann[2], Ryan C. Scott[3], Aaron B. Wilson[1], Maria P. Cadeddu[4], David H. Bromwich[1,5], Johannes Verlinde[6], Dan Lubin[3], Lynn M. Russell[3], Colin Jenkinson[7], Heath H. Powers[8], Maciej Ryczek[8], Gregory Stone[7] & Jonathan D. Wille[1]

Over the past two decades the primary driver of mass loss from the West Antarctic Ice Sheet (WAIS) has been warm ocean water underneath coastal ice shelves, not a warmer atmosphere. Yet, surface melt occurs sporadically over low-lying areas of the WAIS and is not fully understood. Here we report on an episode of extensive and prolonged surface melting observed in the Ross Sea sector of the WAIS in January 2016. A comprehensive cloud and radiation experiment at the WAIS ice divide, downwind of the melt region, provided detailed insight into the physical processes at play during the event. The unusual extent and duration of the melting are linked to strong and sustained advection of warm marine air toward the area, likely favoured by the concurrent strong El Niño event. The increase in the number of extreme El Niño events projected for the twenty-first century could expose the WAIS to more frequent major melt events.

[1] Byrd Polar and Climate Research Center, The Ohio State University, 1090 Carmack Road, Columbus, Ohio 43210, USA. [2] Brookhaven National Laboratory, Building 490D, PO Box 5000, Upton, New York 11973, USA. [3] Scripps Institution of Oceanography, University of California, San Diego, 9500 Gilman Drive, La Jolla, California 92093, USA. [4] Argonne National Laboratory, 9700 South Cass Avenue, EVS/Building 240, Argonne, Illinois 60439, USA. [5] Department of Geography, The Ohio State University, 1036 Derby Hall, 154 North Oval Mall, Columbus, Ohio 43210, USA. [6] Department of Meteorology and Atmospheric Science, The Pennsylvania State University, 605A Walker Building, University Park, Pennsylvania 16802, USA. [7] Australian Bureau of Meteorology, GPO Box 1289, Melbourne, Victoria 3001, Australia. [8] Los Alamos National Laboratory, PO Box 1663, Los Alamos, New Mexico 87545, USA. Correspondence and requests for materials should be addressed to J.P.N. (email: nicolas.7@osu.edu).

Episodes of widespread summer melt have been sporadic in West Antarctica since the phenomenon started being monitored from space in the late 1970s (refs 1,2). Their infrequent occurrence and a lack of robust field measurements to supplement satellite observations leave these melt events insufficiently understood. However, both the geography and climate of West Antarctica conspire to make such events more likely to occur under relatively modest atmospheric warming. Indeed, by virtue of relatively low elevations and frequent intrusions of warm (and moist) marine air[3], West Antarctica experiences a milder climate than neighboring East Antarctica. At the peak of austral summer (December–January), it is relatively common for surface melt to occur over the fringe of ice shelves bordering the Amundsen Sea[2,4], and for surface temperatures over low-lying inland areas to approach 0 °C (ref. 5). In addition, the West Antarctic climate is subject to the influence of large-scale modes of climate variability such as the Southern Annular Mode (SAM) and the El Niño Southern Oscillation (ENSO)[6–10]. These modes and their mutual interactions are responsible for important disruptions of the regional atmospheric circulation that can sustain warm air advection towards the continent for extended periods[3,6].

Here we document a prominent surface melt event that occurred in January 2016 and affected a large portion of the Ross Ice Shelf. This event happened while an important field campaign, the Atmospheric Radiation Measurement West Antarctic Radiation Experiment (AWARE), was ongoing in central West Antarctica. The observations collected during this campaign provided unique insight into some of the physical mechanisms governing surface melting in this otherwise data-sparse region. In particular, these observations highlighted the presence of low-level liquid-water clouds, which may have aided the radiative heating of the snow surface. Furthermore, we explore the large-scale atmospheric factors behind the melt event, namely the role played by the strong 2015–2016 El Niño event and the positive SAM. Building on existing literature and new idealized climate model simulations, we show that the El Niño event is likely responsible for setting up the atmospheric circulation pattern that steered warm air towards the Ross Ice Shelf. We also show that the positive SAM counteracted to some extent the El Niño influence and thus likely mitigated the overall magnitude of the melt event.

## Results

**Melt event captured by satellite and surface observations.** Passive microwave satellite observations (Fig. 1a) indicate that surface melt occurred during one or more days over a broad sector of West Antarctica (termed Ross sector hereafter) in January 2016, with up to 15 melt days over parts of the eastern Ross Ice Shelf and Siple Coast. We assess the significance of this event in the context of the entire satellite record (1978–2016) using two common melt indicators[1]: the melt extent (area of all grid cells with at least one day of melting) and the melt index (MI) (melt area weighted by duration of the melting), both calculated for the Ross sector (red outline in Fig. 1a inset). Bearing in mind that the results are sensitive to the choice of indicator and melt algorithm[2], we estimate that January 2016 was one of the three largest melt events in the Ross sector since 1978 (second behind 1991–92 for MI, and a virtual tie for first with January 2005 for melt extent).

The satellite observations were corroborated on the ground by a number of automatic weather stations (AWSs) that recorded near-surface temperatures near or above 0 °C for several consecutive days during 10–21 January (Fig. 1c). The onset of the melt event on 10 January was accompanied by an abrupt

temperature increase at WAIS Divide and Byrd, in central West Antarctica. The temperature time series from these two sites highlight roughly two phases: Phase 1 (10–14 January), during which the temperatures were at their warmest; and Phase 2 (15–21 January), during which the temperatures gradually decreased towards their pre-event levels. The transition from Phase 1 to Phase 2 is characterized by a shift of the melt pattern towards the Transantarctic Mountains apparent in the AWS temperature time series and in the sequence of daily melt maps (Supplementary Fig. 1).

The January 2016 melt event also coincided with the AWARE field campaign[11], during which comprehensive upper-air, cloud and surface radiation observations were carried out at the WAIS Divide Field Camp (star symbol in Fig. 2a). This site was just downwind and upslope (1,801 m above sea level) of the main melting region and was thus exposed to some of the same weather conditions, as evidenced by the large-scale atmospheric circulation pattern during the melt event (see Fig. 2a and results section 'Regional atmospheric circulation'). The AWARE campaign was also notable in and of itself for providing the first routine upper-air observations from West Antarctica since 1967, when the radiosonde program ended at Byrd Station.

**Cloud and radiative processes.** During the short Antarctic summer, strong onshore winds may by themselves raise the ice sheet's surface temperature ($T_s$) up to the melting point (through exchange of sensible heat), especially at low elevations. However, $T_s$ is ultimately controlled by the full surface energy budget (SEB), being the net of radiative (short- and longwave) and turbulent (sensible and latent) heat fluxes. Clouds exert an important influence on the SEB by modulating the radiative fluxes[12–14], primarily by enhancing downwelling longwave radiation and attenuating incoming solar radiation. In particular, low-level liquid-bearing clouds can have a determinant role in either causing or prolonging melting conditions over ice sheets[15,16].

Model estimates from the ERA-Interim Reanalysis and satellite-based cloud phase retrievals from the Cloud-Aerosol Lidar and Infrared Pathfinder Satellite Observation (CALIPSO) mission for 12 January 2016 (Fig. 2 and Supplementary Fig. 2 and Supplementary Fig. 3) indicate that liquid-bearing clouds were widespread over West Antarctica during the early stage of the melt event. Note that part of the differences between ERA-Interim (Fig. 2a) and CALIPSO (Fig. 2b,c), such as over portions of the Ross Ice Shelf, can be ascribed to complete attenuation of the CALIPSO lidar signal through thick upper-level ice cloud layers. The presence of warm (that is, liquid-bearing) low-level clouds over the Ross Ice Shelf is also apparent in Moderate Resolution Imaging Spectroradiometer (MODIS) observations from 11 January 2016 (Supplementary Fig. 4). The close match between the pattern of ERA-Interim cloud liquid water path (CLWP) and the contours of the melt area on the Ross Ice Shelf highlights the potentially important role of this type of cloud in maintaining melt-prone conditions (compare Supplementary Fig. 1, Supplementary Fig. 2, and Supplementary Fig. 3 for 11–12 January 2016). In addition, the tongue of CLWP stretching from the eastern Ross Ice Shelf to the region of WAIS Divide in Fig. 2a further indicates that AWARE observations may provide insight into the cloud microphysical properties at lower elevation.

The radiosonde profiles (Fig. 3a,b) from AWARE at WAIS Divide captured the large and vertically deep temperature and moisture perturbations associated with the marine air intrusion on 10–13 January. Micropulse lidar measurements (Fig. 3d,e) yielded periods of high attenuated backscatter (>10 dB) and low depolarization ratios (<10%) below 1 km, indicating high cloud

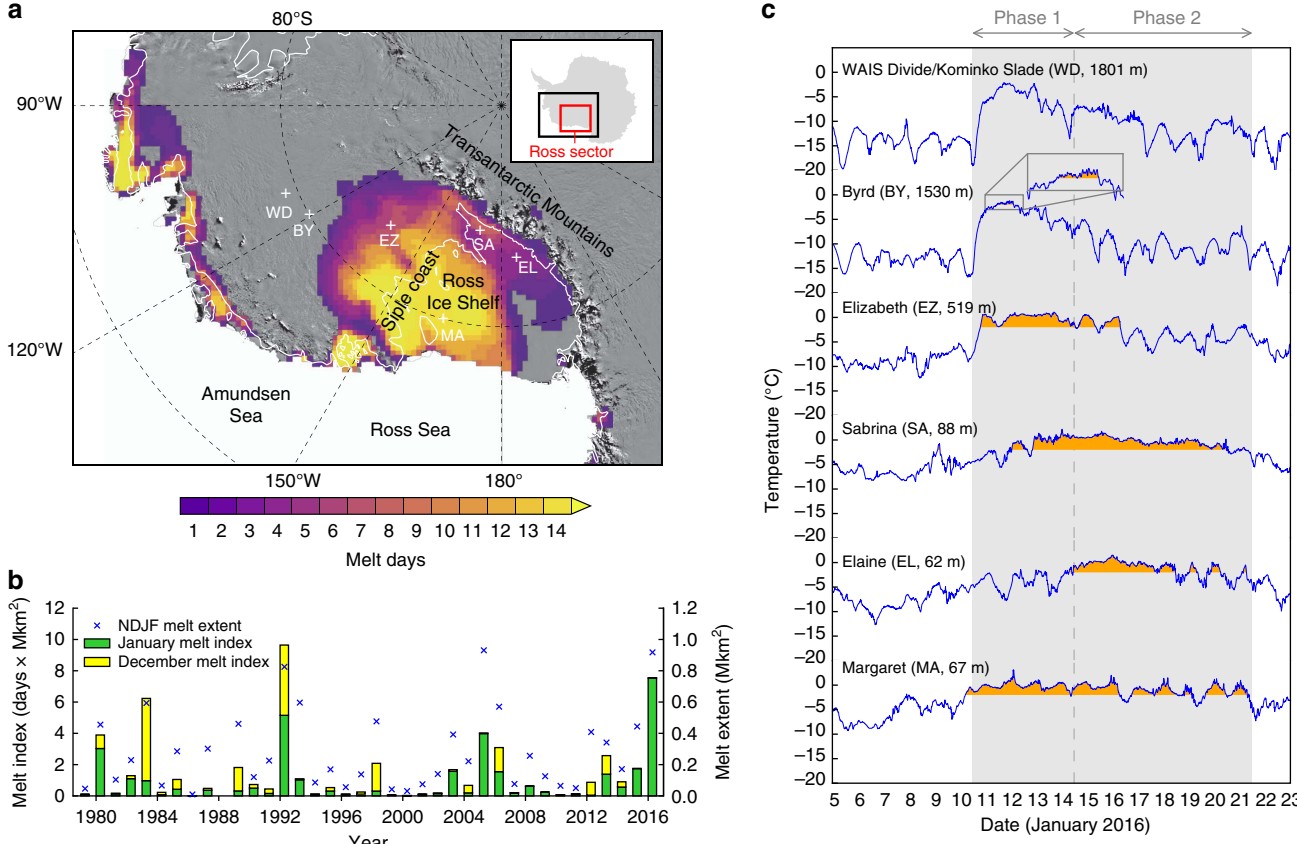

**Figure 1 | The January 2016 melt event captured by satellite and surface observations.** (**a**) Map of West Antarctica showing the number of melt days in January 2016 estimated from passive microwave satellite observations overlaid on a MODIS mosaic image[62]. The white crosses denote the locations of the automatic weather stations (AWSs) shown in **c**. The inset map outlines the boundaries of the background MODIS image (black line) and Ross sector (red line). (**b**) Time series of December and January melt index (bars) and November–February melt extent (blue crosses) calculated for the Ross Sector (see inset map in **a**) and estimated from satellite-based daily melt data. The year refers to January (for example, the 1992 melt indices are December 1991 and January 1992). No data are shown for 1988 owing to insufficient observations. (**c**) Time series of 10-minute near-surface temperatures from six West Antarctic AWSs whose locations are shown in **a**. AWS name abbreviation and elevation above sea level are given in parentheses. Orange shading highlights temperatures above $-2\,^{\circ}\mathrm{C}$ (surface melting can occur despite below-freezing near-surface temperature because of radiative heating).

liquid water content and low ice water content[17] (mixed-phase clouds). The observed CLWP (Fig. 3c) was frequently within $10$–$40\,\mathrm{g\,m^{-2}}$, that is, the range where the cloud radiative enhancement effect previously observed over Greenland[15] occurs. In this range, the clouds are thick enough to enhance the downwelling longwave radiation (Fig. 4a) but thin enough to also allow shortwave radiation to reach the surface (Fig. 4b). The CLWP was within this range 30–40% of the time during 10–13 January, suggesting that this enhancement mechanism contributed to the melt event. This is further supported by the frequent and widespread occurrence of clouds with CLWP within $10$–$40\,\mathrm{g\,m^{-2}}$ simulated by ERA-Interim during the same period (Supplementary Fig. 3 and Supplementary Fig. 5). However, we also notice a significant frequency of CLWP $> 40\,\mathrm{g\,m^{-2}}$ (Fig. 3c), under which shortwave flux is attenuated and longwave flux is similar to blackbody radiation at the cloud effective temperature. These optically thicker clouds represent a contrast to the Greenland cloud radiative enhancement effect in that they signify a more prominent role of thermal blanketing as a consequence of the warm air advection (Fig. 3a,b). The total SEB (Fig. 4e) shows a marked increase in the net energy input into the snowpack (up to $40\,\mathrm{W\,m^{-2}}$), mainly attributable to enhanced downwelling longwave radiation (Fig. 4a,c). This additional energy input is also apparent in the satellite brightness temperatures (Fig. 4e).

**Regional atmospheric circulation.** We trace the immediate causes of the melt event to the presence of an amplified high-pressure ridge (blocking high) over the 90–120°W sector of the Southern Ocean (Fig. 5a–c). By creating a prominent dent in the circumpolar westerly flow, this ridge generated a strong north-south advection of warm marine air towards West Antarctica. The ridge was strongest during 10–13 January (Phase 1) but persisted through 20 January (Fig. 5c), maintaining warm conditions favourable to surface melt in the Ross sector (Phase 2). Positive sea surface temperature (SST) anomalies of $> 2\,^{\circ}\mathrm{C}$ near 50°S, 120°W (Fig. 5b) may have also provided additional heat to the air travelling south (note that the positive geopotential height anomalies near 60°S, 90°W favour anticlockwise motion). Data from ERA-Interim suggest that rain fell over parts of the Ross Ice Shelf at the beginning of the event (Supplementary Fig. 6), which may have preconditioned the snow surface for prolonged melting[18]. Although the reanalysis data should be treated with caution, drizzle was observed at WAIS Divide on 11 January (Supplementary Fig. 7) and rain was witnessed by one field party on the Kamb Ice Stream (black triangle in Fig. 2a) on 12 January (Dr Huw Horgan, Victoria University of Wellington, personal communication).

**Large-scale atmospheric context.** On a broader scale, the melt event occurred during one of the strongest El Niño events on

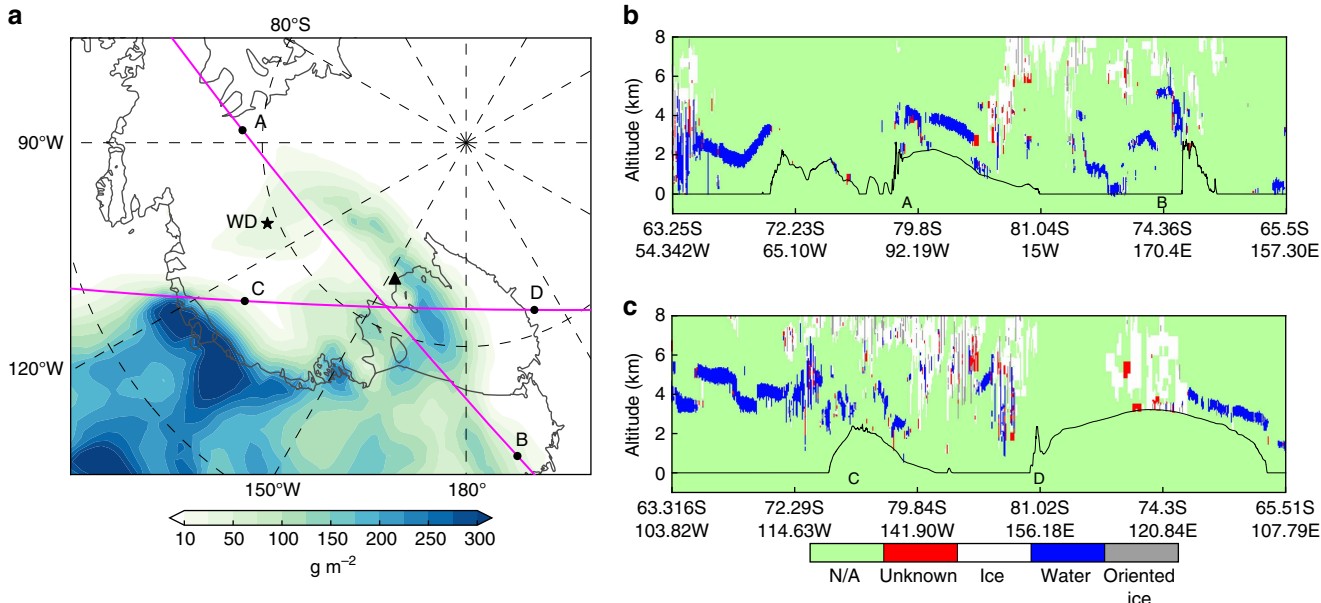

**Figure 2 | Cloud liquid water simulated by ERA-Interim and detected by CALIPSO lidar on 12 January 2016.** (**a**) Map of cloud liquid water path (CLWP) simulated by ERA-Interim for 06:00 UTC on 12 January 2016. The black star denotes the location of WAIS Divide. The two pink lines represent the CALIPSO satellite trajectories corresponding to the vertical profiles shown in **b,c**. The points labelled A, B, C and D along these trajectories are identical to those shown at the bottom of the vertical profiles. The black triangle denotes where rain was witnessed by a field party on 12 January (Dr Huw Horgan, Victoria University of Wellington, personal communication). (**b,c**) Vertical profiles of ice/water phase retrievals from CALIPSO Lidar Level 2 data products over West Antarctica for two time windows on 12 January 2016 (4:15–4:29 UTC in **b** and 7:33–7:46 UTC in **c**). The thin black line represents the surface topography.

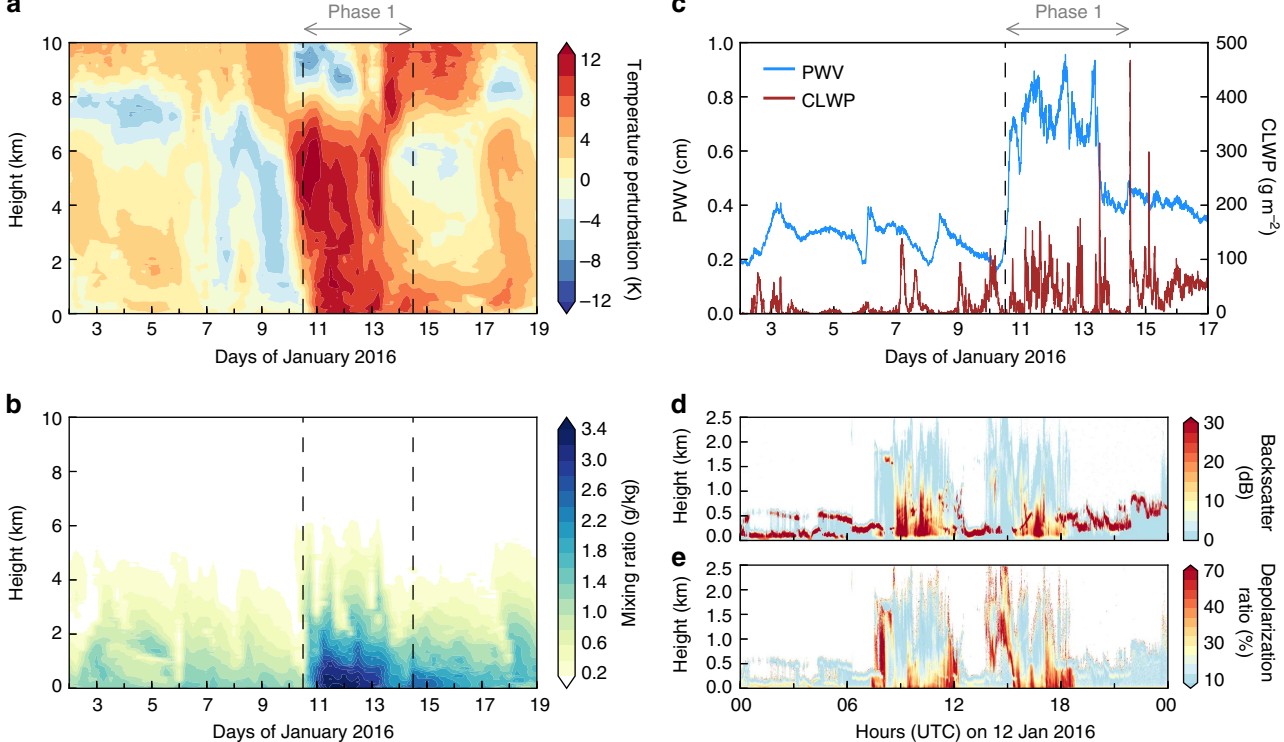

**Figure 3 | Cloud, moisture and temperature observations at WAIS Divide during January 2016.** (**a,b**) Vertical profiles of temperature perturbation and water vapour mixing ratio based on 6-hourly radiosoundings. The temperature perturbation is calculated as the difference between the measured temperature and the mean temperature during the period spanning 4 December 2015 to 18 January 2016 (duration of the field campaign). (**c**) Precipitable water vapour (PWV) and cloud liquid water path (CLWP) estimated from microwave radiometer measurements. (**d,e**) Vertical profiles of attenuated backscatter and depolarization ratio based on micropulse lidar measurements from 12 January 2016. For all vertical profiles, the height is with respect to the ground level (1,801 m above sea level).

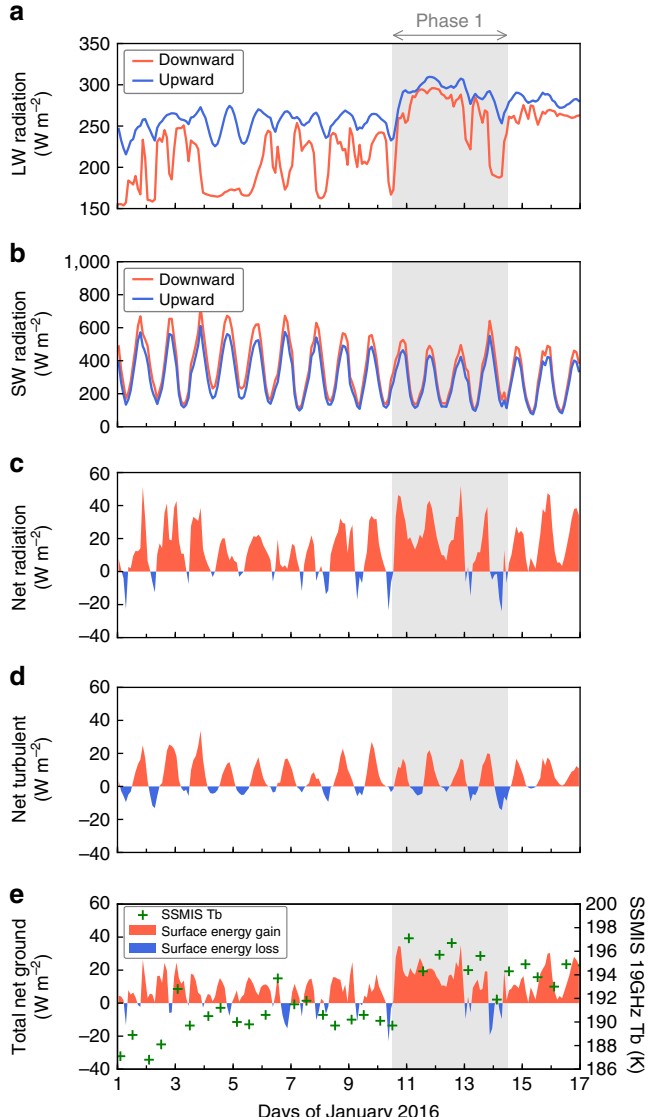

**Figure 4 | Surface energy budget at WAIS Divide in January 2016.**
(**a**) Downward (LW↓) and upward (LW↑) longwave radiation fluxes.
(**b**) Downward (LW↓) and upward (LW↑) shortwave radiation fluxes.
(**c**) Net radiation flux (LW↓ – LW↑ + SW↓ – SW↑). (**d**) Net turbulent flux,
calculated as the sum of sensible (SHF) and latent (LHF) fluxes, where
positive is energy transfer away from the surface to the atmosphere.
(**e**) Total net energy flux into the ground (snowpack) calculated as net
radiation minus net turbulent fluxes. The green crosses represent the
SSMIS 19 GHz horizontally polarized brightness temperatures measured
from space at the location of WAIS Divide.

record (Figs 5b and 6b). Positive (anticyclonic) geopotential
height anomalies in the South Pacific, such as those observed
in January 2016 (Fig. 5b), are a typical signature of El Niño
teleconnections, as seen both in observations[6,19] and climate
model simulations[20,21]. This type of atmospheric pattern
promotes warm air advection to the Ross sector[6], which
explains why surface melt in this area tends to be associated
with El Niño-like conditions[22]. For instance, the prominent melt
events of December 1982 and December 1991–January 1992 (see
Fig. 1b) both occurred in conjunction with El Niño conditions,
characterized by a negative Equatorial Southern Oscillation
Index (SOI) in Fig. 6b. Conceptually, the fact that a circulation
pattern resembling the El Niño teleconnection was present during the

melt event and that this pattern is favourable to warm conditions
over the Ross Ice Shelf points towards a causal link between the
2015–2016 El Niño and the January 2016 melt event. However, in
practice (based on observations available since 1979), the
relationship between the two phenomena remains complex
(see Fig. 7 and related discussion below).

The SAM, which characterizes the strength of the westerly
winds around Antarctica, is an important modulator of the
tropical influence in the South Pacific sector of the Southern
Ocean[7]. During November 2015–January 2016, the SAM
Index remained predominantly positive (Fig. 6a), indicating
stronger-than-normal westerly winds. This is reflected in the
negative geopotential height anomalies over Antarctica in Fig. 5b.
The conjunction of a strong El Niño and a positive SAM phase
was unusual since, in austral summer, the latter is most often
associated with La Niña-like conditions[8,23] (conversely, a negative
SAM phase most often occurs in conjunction with El Niño-like
conditions). Furthermore, a positive SAM phase hampers
meridional heat exchange between middle and high latitudes[7],
and is thus generally not conducive to surface melt in West
Antarctica[22]. Thus, if anything, the positive SAM phase that
prevailed before and during the melt event should have favoured
colder-than-normal, not warmer-than-normal, conditions in
West Antarctica.

**Contribution of El Niño and the SAM.** Properly understanding
the mechanisms responsible for the January 2016 melt event
requires investigating the potential roles of El Niño and the SAM.
This in turn can provide insight into the recurrence of such event
in the future (see Discussion). Figure 7 provides a means to
visualize the three-way relationships between West Antarctic
summer melt, the SAM, and the ENSO phenomenon (repre-
sented by the Equatorial SOI) since 1979. Note that, in this figure,
the two melt indices are calculated for December–January, and
the two climate indices are November–January averages. Figure 7
shows that, in general, less melt tends to occur during La
Niña-like conditions (SOI > 0) and a positive SAM phase,
whereas more melt tends to occur during El Niño-like conditions
(SOI < 0) and a negative SAM phase. This qualitative assessment
is confirmed by tallying positive and negative melt anomalies
depending on the sign of the two climate indices (Supplementary
Table 1). It is worth noting that the directions of the relationships
are similar to those previously found between Antarctic-wide
melt and the SAM Index and SOI[22].

These relationships are by no means simple. For example,
not all major El Niño events are accompanied by widespread
surface melt in West Antarctica (for example, 1997–98); not
all prominent West Antarctic melt events coincide with strong
El Niño events (for example, 2005); and the magnitude of
West Antarctic melt does not scale with the intensity of El Niño
events. Accordingly, it is not possible to establish with certainty
whether the 2015–2016 El Niño caused (in a deterministic sense)
the January 2016 melt event, a problem inherent to weather
and climate phenomena[24]. It is not uncommon for the polar
jet around Antarctica to exhibit large meanders, giving rise to
warm marine air intrusions[3], even in the absence of an El Niño
event. Following a probabilistic approach, we seek rather to assess
the likelihood of the January 2016 melt event occurring given
the concurrent strong El Niño and positive SAM conditions.
The statistically small number of El Niño events (especially of
strong events such as 1982–83, 1997–98 and 2015–16) observed
since 1979 does not permit robust statistical analysis. However,
climate model simulations can alleviate this issue by generating
larger samples of events.

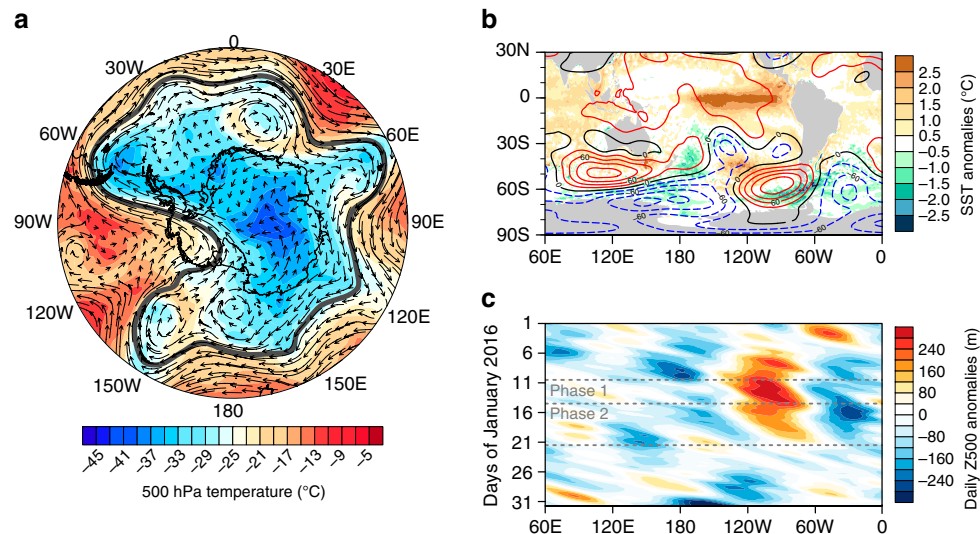

**Figure 5 | Atmospheric circulation associated with the January 2016 melt event.** (**a**) Map showing the air temperature (shaded) and wind vectors at 500 hPa at 00 UTC on 12 January 2016 based on ERA-Interim. The thick black line outlines the 5,300 meter geopotential height contour separating cold polar air masses from warmer mid-latitudes air masses. (**b**) Sea surface temperature anomalies (shaded) overlaid with ERA-Interim 500 hPa geopotential height (Z500) anomalies (contour lines) in January 2016. Solid red, dashed blue and solid black contour lines represent (respectively) positive, negative and zero Z500 anomalies. Anomalies are calculated with respect to the 1971–2000 period. (**c**) Hovmöller diagram showing daily meridionally averaged Z500 anomalies within latitudes 50–80°S during January 2016. The anomalies are with respect to the 1979–2016 January monthly mean.

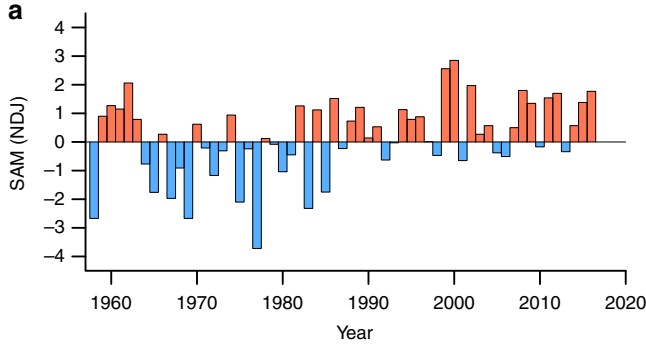

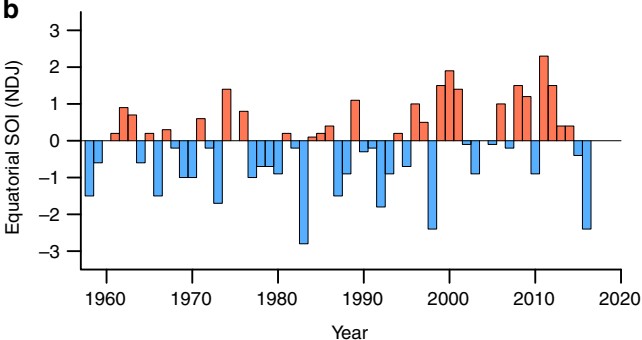

**Figure 6 | Time series of November-January averaged climate indices.** (**a**) Southern Annular Mode Index. (**b**) Equatorial Southern Oscillation Index. The series span 1957–2016. The year corresponds to the month of January. The last data point is for November 2015-January 2016.

**Modelling experiment set-up.** We used the Community Atmosphere Model (CAM) version 4 to conduct a set of four 15-year idealized simulations (see Methods for details). Three simulations were forced with annually repeating SSTs characteristic of past strong El Niño events, representing a total of 45 El Niño events. The fourth simulation was forced with

annually repeating climatological SSTs to serve as control run. A model SAM index was calculated for each simulation based on principal component analysis of Southern Hemisphere monthly 500 hPa geopotential height anomalies. Here again, we considered the average SAM Index for November–January.

Estimating surface melt occurrence in the CAM model can be problematic as the process is affected by model grid resolution, model temperature biases and model deficiencies in the placement of key atmospheric features[25–28]. To circumvent these issues, we used anomalies in the model near-surface air temperature as an indicator of melt-prone conditions. We calculated these anomalies for each simulation by subtracting the long-term monthly means of the control simulation, and considered the mean anomalies for December–January spatially averaged over a broad Ross sector of West Antarctica (75°–90°S; 180°–90°W). We labelled these anomalies warm events or cold events depending on their sign.

**Results from model simulations.** Based on the model SAM Index and series of warm and cold events, we generated a contingency table (Table 1) tallying the number of events per type (warm or cold) and phase of SAM (positive, negative or neutral) across all simulated major El Niño events. Out of 45 El Niños, warm events occur 32 times (71.1%) versus 13 times (28.9%) for cold events. This result is consistent with the known positive impact of the El Niño teleconnection pattern in the South Pacific on West Antarctic temperatures already mentioned[6]. Out of the 32 warm events, 15 (46.9%) occur during a negative SAM phase. Out of the 13 cold events, 8 (61.5%) occur during a positive SAM phase. The chi-square statistic is significant at $P < 0.01$, meaning that the type of event is significantly dependent on the combined states of El Niño and SAM. Such dependence confirms findings from previous literature[7,29,30].

Expanding on the previous analysis, we also find that 9 out of the top 10 warmest events occur during a negative or neutral SAM phase, while 9 out of the top 10 coldest events occur during a positive or neutral SAM phase. This demonstrates that the Ross sector is much more likely to experience conditions favourable to

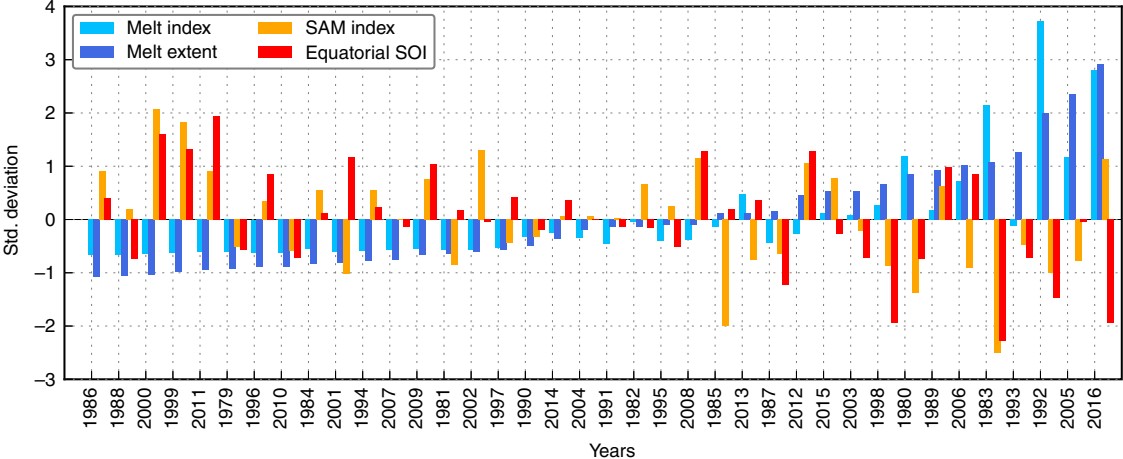

**Figure 7 | 1980–2016 standardized SAM Index, Equatorial SOI, West Antarctic melt index and melt extent anomalies.** Years are rank-ordered from lowest to highest melt extent. For each year (x axis), the plot displays (in the following order) the melt index (light blue), the melt extent (dark blue), the SAM Index (orange), and the equatorial SOI (red). The plot is adapted from ref. 22.

**Table 1 | Contingency table tallying the count of December–January warm and cold events as a function of the phase of the November–January Southern Annular Mode in idealized CAM4 model simulations.**

| | Type of temperature anomaly* | | |
|---|---|---|---|
| | **Warm** | **Cold** | **Row totals** |
| +SAM[†] | 3 (7.82)[‡] | 8 (3.18)[‡] | 11 |
| Neutral SAM[†] | 14 (13.51)[‡] | 5 (5.49)[‡] | 19 |
| −SAM[†] | 15 (10.67)[‡] | 0 (4.33)[‡] | 15 |
| Column totals | 32 | 13 | 45 |

*The temperature anomalies are December–January means, spatially averaged over the 75°–90°S, 180°–90°W sector of West Antarctica, and calculated with respect to the 15-year mean of the control simulation. The warm and cold columns correspond to positive and negative temperature anomalies, respectively.
†The SAM index represents the November–January average. The positive (+SAM), neutral, and negative (−SAM) phases are defined based on the ±0.5 standard deviation of the SAM index.
‡Expected counts (which assume independence between the two variables) are given in parentheses. The chi-square statistic is 14.67. The P value is 0.00065.

surface melting during El Niño events when a negative or neutral SAM is present. Given that the SAM was in a predominantly positive phase before and during the January 2016 melt event, our model results suggest that the state of the SAM likely mitigated the magnitude (areal extent, duration) of surface melt in West Antarctica during the austral summer of 2015–2016. In other words, the 2015–2016 melt season would likely have been more prominent had the SAM been in a negative or neutral phase, more commonly associated with El Niño events.

### Discussion

Further research is needed to better understand the various mechanisms behind major West Antarctic melt events and to accurately predict their future occurrence. Accurate prediction is contingent on the ability of climate models to resolve the broad range of factors responsible for these events, from the large-scale climate drivers to the regional atmospheric circulation to the microphysical and radiative processes. Among them, the simulation of ENSO and its teleconnections, and the representation of high-latitude mixed-phase clouds (such as those observed at WAIS Divide in January 2016) are two key areas in need of improvement in climate models[15,31,32].

Future changes in the intensity of ENSO events are currently estimated with greater confidence than ENSO-related long-distance climate changes, owing in large part to model difficulties in simulating the present-day mean state climate[32,33]. The frequency of extreme El Niño events is projected to increase over the course of the twenty-first century[32,34]. Given the role of El Niño-related atmospheric circulation in promoting warm air advection to the Ross sector, a greater number of extreme El Niño events could foster more frequent major melt events in this area. One source of uncertainty lies in the modulating effect of the SAM on this teleconnection. This effect has become less clear after January 2016. Indeed, the conjunction of a strong El Niño/strong teleconnection on the one hand, and strong westerlies on the other hand was at odds with known tropical-high latitude interactions[7,8,10,22]. Yet, this scenario may grow more likely in the future[35] as anthropogenic forcings are expected to continue favouring positive SAM conditions in austral summer[36].

Finally, the January 2016 melt event demonstrates that the present-day climate of West Antarctica already allows for extensive surface melt to occur occasionally. In this regard, two recent modelling studies[37,38] have come to rather different conclusions about the future evolution of surface melt over the Ross Ice Shelf and its impact on the WAIS mass balance. One study[37] suggests that the phenomenon will remain minimal throughout the twenty-first century, and is, therefore, unlikely to contribute significantly to the destabilization of the WAIS. The other study[38] projects that the Ross Ice Shelf will experience extensive surface melt and retreat substantially by 2100, thereby accelerating the disintegration of the WAIS. In this context, the extent to which the January 2016 event is a precursor of the climate of West Antarctica in the coming decades is uncertain. But our study highlights some of the key mechanisms that need to be resolved to address this question.

### Methods

**Satellite-based melt data.** Surface melt over ice sheets can be easily detected from space as the appearance of liquid water in the snowpack causes a sharp increase in microwave brightness temperature[2]. Here we estimated surface melt occurrence using daily satellite brightness temperature (Tb) data obtained from the National Snow and Ice Data Center. The data consisted of twice-daily observations (from ascending and descending satellite passes) from the following sensors: The Scanning Microwave Multichannel Radiometer (SMMR) onboard the Nimbus-7 satellite (1978–1987); the Special Sensor Microwave/Imager (SSM/I) onboard the Defense Meteorological Satellite Program (DMSP) F-8, F-11, and F13 satellites (1987–2009); and the Special Sensor Microwave Imager Sounder (SSMIS) onboard the DMSP F-17 satellite (2006–present). We used horizontally polarized Tb data in the K-band (18 GHz for SMMR, 19 GHz for SSM/I–SSMIS), commonly used for

melt detection over ice sheets[1,2,39]. The data were provided on National Snow and Ice Data Center's Southern Hemisphere EASE-Grid with 25 × 25 km grid cells. We filled the gaps in the SMMR data (available only every other day) by linearly interpolating the data from the two adjacent days. We filled the gaps in the SSM/I–SSMIS data only if they did not exceed one day. To ensure consistency between the different sensors, we adjusted all SMMR and SSM/I Tb data to SSMIS F-17 using the regression coefficients derived by refs 40–42. The only exception was for the adjustment between SSM/I F-13 and SSMIS F-17, for which we derived our own coefficients (see Supplementary Fig. 8). All coefficients used in our adjustment procedure (along with their references) are listed in Supplementary Tables 2 and 3.

For a given grid cell and a given day, we determined that melt was occurring as soon as one of the two daily Tb observations exceeded a threshold value ($Tb_{melt}$) defined as $Tb_{melt} = Tb_{ref} + \Delta T$, where $\Delta T = 30$ K and $Tb_{ref}$ is a reference temperature. $Tb_{ref}$ was calculated as the 12-month average from 1 April–31 March after filtering out all melt days as in ref. 43. When $Tb_{ref}$ could not be calculated (for example, at the beginning/end of a satellite record), we used the $Tb_{ref}$ value from the previous or following year (whichever matched the sensor/satellite). This overall algorithm was shown to be particularly well suited for detecting melt in dry-snow areas[2], such as found in the West Antarctic interior. The MI shown in Fig. 1b was calculated as follows:

$$MI = A \cdot \sum_{i=1}^{N} m_i$$

Here, $A$ is the area of a pixel, $m_i$ is the number of melt days during a month for pixel $i$, and $N$ is the number of pixels inside the Ross sector (red outline in Fig. 1a).

**Observations from West Antarctic Ice Sheet Divide.** The 2015–2016 AWARE field campaign ran from 4 December 2015 through 18 January 2016, and deployed ARM Mobile Facility instruments[44] at WAIS Divide. Estimates of upper-air temperature and moisture were obtained from six-hourly rawinsonde launches[45] and continuous retrievals using a profiling microwave radiometer (MWR)[46,47]. A micropulse lidar[48,49] measured cloud layer elevation and thermodynamic phase using both direct and cross-polarized laser returns. Column-integrated precipitable water vapour and CLWP were retrieved using the combined data from the profiling MWR and a two-channel MWR[46,50]. Upwelling shortwave and longwave radiative flux components were measured by a Surface Energy Balance system[51]. Downwelling flux components were measured by a Sky Radiation System[52], which consists of a normal incidence pyrheliometer and shaded pyranometers and pyrgeometers. The global downwelling shortwave flux ($S_{down}$) was computed as follows:

$$S_{down} = S_{dir} \cos\theta_z + S_{diff}$$

Here, $S_{dir}$ is the direct solar beam from the normal incidence pyrheliometer, $\theta_z$ is the solar zenith angle and $S_{diff}$ is the diffuse flux from a shaded pyranometer. Surface latent and sensible heat fluxes were derived using the algorithm of ref. 53 and surface measurements of temperature, moisture and wind speed from the ARM surface meteorological instrumentation[54]. The velocity roughness length used in the algorithm was derived for the WAIS site using momentum fluxes from an Eddy Correlation Flux Measurement System[55].

**Model simulations.** A basic description of our modelling experiment is already provided in the main text for clarity. A few additional details are given here. We conducted four 15-year simulations using the CAM version 4 (ref. 56) with sea-surface conditions specified as in refs 10,20. Three simulations were forced with cyclic, annually repeating 12-month global SSTs based, respectively, on the major El Niño events of 1982–83, 1997–98, and a scaled composite of other historical El Niño events (see details in ref. 20). The fourth simulation (used as control) was forced with annually repeating SSTs based on climatological monthly mean SSTs for the period 1981–2010. The SAM was defined in each simulation using the first principal component of monthly mean 500 hPa geopotential height anomalies poleward of 10°S (these anomalies were obtained by removing the 15-year monthly means of the control simulation).

**CALIPSO cloud particle phase.** The vertical profiles of cloud particle phase shown in Fig. 2b,c are based on measurements from the Cloud-Aerosol Lidar with Orthogonal Polarization (CALIOP) flying onboard the CALIPSO satellite. Cloud phase retrievals are part of CALIPSO Lidar Level 2 Vertical Feature Mask (VFM) products (version 3.30) available from NASA's Atmospheric Data Center (https://eosweb.larc.nasa.gov/clouds). Details about the cloud phase discrimination algorithm are given in ref. 57. Technical documentation about the VFM products is provided in ref. 58. Regions denoted as unknown in Fig. 2b,c are where cloud phase determination is ambiguous. In this regard, note that the algorithm used for VFM version 3 products does not attempt to identify mixed-phase clouds. Regions denoted as N/A (not applicable) are classified as clear air in the VFM products. These are regions where no features (cloud or aerosols) are detected either because none are present or because of lidar backscatter signal attenuation through overlying cloud layers.

**Other data.** The AWS temperature estimates used in Fig. 1c are 10-minute data obtained from the Antarctic Meteorological Research Center at the University of Wisconsin-Madison (ftp://amrc.ssec.wisc.edu/pub/aws/10min/rdr/). On these stations, the temperature sensor is at a height of three meters above the surface. The CLWP data used in Fig. 2a and the temperature, geopotential height, and wind data in Fig. 5a–c are from the ERA-Interim Reanalysis[59] (http://apps.ecmwf.int/datasets/). Note that CLWP corresponds to ERA-Interim total cloud liquid water field. The SST data used in Fig. 5b are from NOAA's Optimum Interpolation 1/4° Degree Daily Sea Surface Temperature Analysis, Version 2 (ref. 60; https://www.ncdc.noaa.gov/oisst). The SAM Index[61] used in Fig. 6a is courtesy of G.J. Marshall (https://legacy.bas.ac.uk/met/gjma/sam.html). The Equatorial SOI used in Fig. 6b is provided by NOAA's Climate Prediction Center (http://www.cpc.ncep.noaa.gov/data/indices/).

**Code availability.** The source code of the CAM4.0 global atmospheric model can be obtained free of charge through the Community Earth System webpage (http://www.cesm.ucar.edu/models/ccsm4.0/cam/).

**Data availability.** All observations from the AWARE campaign are available from the ARM Data Discovery website (http://www.archive.arm.gov/discovery/). For quick access, users can enter 'AWARE' in the search box (top-left corner of the page) and click on AWR—AWARE (ARM West Antarctic Radiation Experiment in the search results.

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

## Acknowledgements

J.P.N., A.B.W., and D.H.B. were supported by National Science Foundation (NSF) grants PLR-1443443 and PLR-1341695. A.M.V. is supported by the U.S. Department of Energy (DOE) under Contract DE-SC0012704. R.C.S. was supported by NASA grant NNX15AN45H. M.P.C. is supported by the DOE under Contract DE-AC02-06CH11357. J.V. was supported by NSF grant PLR-1443495. AWARE is supported by the DOE ARM Climate Research Facility and NSF Division of Polar Programs. We thank WAIS Divide Station Manager E. Beazley and her crew for their field support. We thank the ARM data management and data quality personnel (particularly N. Keck, C. Stuart, J. King and A. Theisen) for timely review and delivery of the WAIS data, and instrument mentors (particularly D. Cook and J. Kyrouac) for assistance with the data. CAM model simulations were conducted at the Ohio Supercomputer Center. The University of Wisconsin-Madison Automatic Weather Station Program, provider of the AWS data, is supported by NSF Grant ANT-1245663. This paper is contribution 1567 of the Byrd Polar and Climate Research Center.

## Author contributions

J.P.N. processed the SMMR-SSM/I-SSMIS, CALIPSO and AWS data and some of the ERA-Interim data and climate indices. A.M.V. processed the AMF radiation and energy flux data. R.C.S. was the AWARE Site Scientist at WAIS Divide and processed the MODIS data. A.B.W. conducted the CAM4 model simulations, processed the CAM4 output and some of the ERA-Interim data and climate indices. M.P.C. processed the AMF microwave radiometer data. J.V. processed the AMF micropulse lidar and sonde data. J.P.N., A.B.W., D.H.B., and J.D.W. provided the meteorological analysis. AWARE was conceived and planned in collaboration with ARM and the U.S. Antarctic Program (USAP) by D.L., D.H.B., A.M.V., J.V., and L.M.R. The ARM WAIS Divide field party was C.J., R.C.S., H.H.P., M.R, G.S. and D.L., with C.J. serving as ARM Site Engineer. J.D.W. served as a weather observer for USAP at WAIS Divide. J.P.N. wrote the paper with help from A.B.W., D.H.B., D.L. and A.M.V.

## Additional information

**Competing interests:** The authors declare no competing financial interests.

