## [Peer Review File · Nature Communications]

Reviewers' comments:

Reviewer #1 (Remarks to the Author):

Review of "January 2016 extensive summer melt in West Antarctica favored by strong El Niño" by Nicolas et al., submitted to Nature Communications.

Review Summary:

Nicolas et al document an extensive surface melt event during early 2016 in West Antarctica using a combination of remotely sensed melt observations, in situ ground and atmospheric observations from WAIS Divide, and atmospheric and oceanic reanalyses. Using these datasets, the authors attribute the melting to the strong 2015 El Niño and the strongly positive Southern Annular Mode (SAM) during the 2015-2016 austral summer. Based on these inferred links, the authors conclude that extensive WAIS melting may increase in frequency in the future, due to the expected increases in El Niño intensity and positive phases of the SAM.

Understanding the causes and impacts of extensive melting across WAIS is certainly an important endeavor. The authors do a nice job of documenting the melt event via the passive microwave melt time series and its potential direct causes (i.e., warm air intrusion and low level liquid clouds). However, the link between this event, ENSO, and SAM is not robustly presented. To me, this represents a major flaw in the manuscript. For example, there is no statistical analysis between melt and these climate indices necessary to link the phenomena. To establish a link (which would then better support the rather speculative conclusions about future increases in WAIS melt), a statistical analysis is necessary as well as an investigation of the past extensive WAIS melting episodes. Otherwise, we are simply learning about the weather conditions during the 2016 melt event, not the climatic processes and teleconnections ultimately governing or influencing extensive WAIS melting. The fact that the 2016 event was nearly identical to the 2005 event in terms of extent and that there was a lack of a strong El Niño in 2004-2005 suggests that other important factors beyond El Niño are at play.

I would also suggest toning down statements about how increases in WAIS melt may lead to its future destabilization (e.g., the summary paragraph sentence stating "... the potentially important role that surface melt could play in the disintegration of the WAIS in the coming decades"). While WAIS surface melt events like that in 2016 are anomalous in the satellite record and quite extensive, they are certainly weak relative to melt conditions over Greenland or on the Antarctic Peninsula where surface melt has been linked to hydrofracture.

Specific Comments:

Figure 1:

Please label each AWS on the map.

Line 59-60:

The westward propagation towards the TAM at the onset of "Phase 2" is not very clear in the supplementary figure 1. Perhaps you could indicate the extent of the Ross Ice Shelf to make it more clear where the TAM are in the figure?

Line 65-66:

The author state that a positive SST anomaly around 50S/120W contributed to warm air flowing south, citing figure 2b. I have a few questions here (the first two apply to Figure 2C as well):

- What time period were these anomalies derived from? This needs to be stated.
- Are these anomalies significant? Can you shade areas that are significant?
- The link to the positive SST anomaly is not well supported by the presented analysis, in particular given that a larger and negative SST anomaly exists closer to WAIS and over which this

"warm" air would have needed to travel. Would these not cancel out the much smaller and isolated area of warm water?

Line 67:

How confident are you in the ERA-Interim rainfall data in supplementary figure 2? The rain over WAIS and particularly the rain at 120W and >80S seems very suspect to me. Could this rainfall be corroborated by passive microwave brightness temperatures?

Line 68:

Could the blocking event that set up prior to phase 1 have also preconditioned the snow surface for melting (e.g., by increasing shortwave radiation under clear skies)? The atmospheric blocking prior to the warm air advection seems like it might be an important piece of the puzzle.

Lines 83-85:

It is stated that El Nino-linked blocking and strong westerlies (due to strongly positive SAM) "caused the strong southward warm air advection that contributed to the melt event." This statement of a causal link is not supported by any evidence. Rather, only the occurrence of +SAM and El Nino are established. Furthermore, the meanders in the jet stream around Antarctica (as shown in Fig 2a) are common over short time scales, irrespective of the state of SAM. Thus, in my opinion, the role of SAM (and ENSO) here remains questionable.

Lines 89-91:

Please expand upon how the clouds affect the radiative fluxes for the reader that might not be familiar. What are the most relevant terms here to ice sheet surface melt - shortwave or longwave fluxes, both? Also, please be more explicit in the physical processes influencing melt that relate to low level liquid clouds.

Lines 100-103 / Figure 3d,e:

The plots in figure 3 d,e from 12 January (two days after the start of phase 1) show periods of high backscatter and low depolarization ratio as stated in the text, but they also show the opposite as well. The wording here should be changed accordingly (perhaps state "periods" or "intervals" of high attenuated backscatter...). Perhaps more importantly, the data also suggest non-mixed phase clouds almost as frequently during this interval. Like my comment above regarding Lines 65-66, this seems to be a selective interpretation of the data.

Lines 108-109 / Supplementary Figure 3:

The ERA-Interim data seem to match up nicely with the field observations from WAIS Divide, however, they also show that much of the ice sheet that underwent melting did not experience similar conditions to those at WD. This raises the question of whether the observations at WAIS Divide of liquid clouds are relevant to interpreting the cause of the widespread melt event.

Line 111:

Remove "net" as you are referring only to the downward longwave radiation.

Line 112:

The additional energy input has warmed the surface snow temperature, which is reflected in the satellite brightness temperatures (not the energy input directly as it is stated currently). This raises another point: Did you model the SEB at WAIS Divide to calculate snowpack temperature? Did surface melt occur at WAIS Divide? Based on your excess energy ($\sim >30 \text{ W m}^{-2}$), it seems that the surface would have been very close to melting (depending on the magnitude of heat loss to the subsurface).

Lines 112-114:

This sentence is a bit unclear. What is the "positive perturbation" referring to? From looking at Figure 4e, it seems that there remains both a positive perturbation in both the T_b time series and

the net SEB during and after phase 1. Also, I'd be skeptical about having a heat flux from deeper layers up to the surface at WD - do you have subsurface temperature profiling to support this claim?

Line 123:

"changes in other aspects of the phenomenon" is unclear. Can you be more specific?

Line 144:

Change "passive microwave emission" to "microwave brightness temperature".

Line 157:

Tedesco 2009 does not go into detail on the method as cited here. Did you generate your own regression coefficients, or use those of Jezek et al (1991) and Abdalati et al (1995) as employed by Liu et al (2006)? If not, the methodology used in this study should be better explained and regression coefficients listed in supplementary information - thus allowing reproduction of the dataset.

Reviewer #2 (Remarks to the Author):

Review of paper "January 2016 extensive summer melt in West Antarctica favored by strong El Niño" by Nicolas et al., submitted to Nature Communications.

The authors present an interesting case study of widespread surface melt over Western Antarctica and the Ross Ice Shelf. The authors combine detailed surface-based observations from different Automatic Weather Stations, the AWARE Radiation Experiment site, satellite observations, and reanalysis data to characterize the melt event. The manuscript is well written and easy to follow. The characterization of the synoptic situation that led to the surface melt is in my view the strongest part of the manuscript. Advection of warm air was driven by a strong and persistent high-pressure ridge slightly west of the investigation area. This ridge persisted over several days and allowed for warm air to be advected southwards over the investigation area.

Using the combination of AWARE surface observations and ERA reanalysis the authors also find thin, liquid-bearing clouds to enhance surface melt. The detailed radiation measurements presented in Figure 3 substantiate this point for AWARE for the period 10-13 January but do not provide data for the period after January 19. It would have been interesting to see the measurements also for 'Phase 2' of the melt event (even though during that later time period the event had moved to the east).

In addition, ERA reanalysis (Figures S1 and S2) shows the occurrence of liquid precipitation and widespread liquid clouds over the investigation area. While the findings based on ERA reanalysis certainly are in agreement with the authors claim, it is unclear to me how much one can trust ERA over the southern high latitudes. NWP models are known to underestimate the amount of low-level liquid in southern clouds, in particular in regions of subsidence (Naud et al., 2014). Mid-representation of clouds is also known to lead to large biases in the radiation balance (e.g. Trenberth and Fasullo 2010). Such biases in model-derived cloud parameters could result in either an under- or overestimation of the role of liquid clouds in the context of this study. In order to substantiate the point made by the authors, it would be worthwhile to try augmenting the study with additional satellite data, preferably with MODIS observations of liquid clouds and/or space-borne lidar observations from NASA's CALIOP mission.

The largest issue I have with the manuscript lies in the implied relation between El Niño and the observed warming. I have to admit that I am not an expert in large-scale teleconnections, so my

following comments should be taken with a grain of salt.

In my view, the manuscript does not necessarily justify the authors' claim that the warm air advection arose "from the conjunction of a record El Niño event and strong circumpolar westerly winds". Clearly, both a record El Niño and strong westerly winds existed. However, the relation between El Niño and the warming appears to be not actually found in this study. Rather, the study relies on earlier studies that appear to have shown the effect. See lines 71 - 73 in the manuscript and references 12 and 18 therein. New mechanisms or teleconnections relating El Niño to melting in Antarctica are in my understanding not discussed in the study.

Another confusing issue arises from the lack of consistency between the observed melt events Jan-83, Jan-92, Jan-05, and Jan-16 and the corresponding SAM and ONI anomalies. From Figure 2 we have:

Jan-83: SAM negative, ONI positive

Jan-92: SAM neutral, ONI positive

Jan-05: SAM neutral, ONI slightly positive/neutral

Jan-16: SAM positive, ONI positive

There are also years (e.g. Jan-10) where ONI is slightly positive/neutral and no melt occurs.

I think this generally substantiates the author's own finding that "Further research is needed to better understand [...] mechanisms behind major West Antarctic melt events [...]".

Finally, if my above assessment holds, it would also call in question the speculative last sentence of the introduction where the authors state that surface melt could potentially (sic!) play an important role in the disintegration of the WAIS in the coming decades. It is of course true that "potentially" surface melt can play an important role. However, in my view, the study presented here does not necessarily provide strong evidence one way or another.

Naud, C., J. F. Booth, and A. D. Del Genio, 2014: Evaluation of ERA-Interim and MERRA Cloudiness in the Southern Oceans. *J. Climate*, 27, 2109-2124, doi:10.1175/JCLI-D-13-00432.1.

Trenberth, K. E. and J.T. Fasullo, 2010: Simulation of Present-Day and Twenty-First-Century Energy Budgets of the Southern Oceans, *J. Climate*, DOI: <http://dx.doi.org/10.1175/2009JCLI3152.1>

Reviewer #3 (Remarks to the Author):

Major comments

This paper describes an extensive melt event across West Antarctica that occurred in January 2016. The event occurred during a strong blocking episode that resulted in the advection of warm, maritime air into West Antarctica and the formation of mixed-phased clouds. The authors argue that the combination of relatively warm air and enhanced downward longwave radiation from the cloud cover contributed to the duration and intensity of the surface melting. Further, the blocking episode is described as part of a teleconnection pattern driven by a strong El Niño event in the tropical Pacific. The most unique aspect of this paper is the inclusion of field measurements of surface energy balance and clouds taken during the AWARE campaign. Such measurements are rare and their use in conjunction with satellite data and atmospheric reanalysis makes this paper particularly noteworthy. The paper is very well-written, easy to read, and concise. The weakest link is the climate change connection - is this one event a sign of things to come? Will surface

melting be a significant contributor to ice sheet mass loss? Has this type of melt event occurred in the past? The answers to these questions are difficult but I think are what would make it a strong Nature Communications paper as opposed to a noteworthy field study. If I have one major recommendation to improve the paper, it is to flesh out one of these questions a bit more - perhaps include either some historical context from ice cores or some results from climate models. Since El Nino events are intermittent and it is hard to predict whether a given event will produce the right teleconnection pattern, it is hard to say whether the link of El Ninos to melting episodes is significant. I am wondering if just the slow atmospheric warming associated with climate change would be sufficient (by say 2100) to significantly increase the probability of summer melt events, regardless of what happens to the frequency or intensity of El Ninos? Model results may not be reliable enough to make the link to surface melting directly, but they may be able to shed light on projected warming trends and when the forced climate change signal in surface air temperature will emerge above natural variability, and when this signal plus a strong El Nino would surely produce near-freezing temperatures on WAIS.

Minor comments

Lines 67-68 and Supp figure 2: Is there any evidence that it actually rained on the WAIS during this event, or is it just that the reanalysis model forecasted rain? Were there any ground observers at WAIS Divide to corroborate the forecasted rain? In any event, the rain does not look very extensive compared to the area of melting. I'm not really sure that these data and the argument about preconditioning of the snow add much to the story.

Lines 133-142: I think more explanation is needed here. Why did the two studies reach such different conclusions? Did they use different climate change scenarios (RCP 4.5 vs RCP 8.5), different algorithms to calculate melt, or different ice sheet or climate models? Does the present study shed any light on this issue?

Responses to Reviewers

Response common to all Reviewers

We thank the reviewers for their comments and suggestions. The manuscript has changed substantially since our initial submission. Although some (relatively minor) changes were made to comply with Nature Communications editorial guidelines, the bulk of the revisions was made to address concerns raised about our initial assessment of the contribution of the strong El Niño and the positive SAM to the January 2016 melt event. Since these concerns were shared by the three Reviewers, we begin with a response addressed to all Reviewers before providing more detailed responses to each Reviewer's comments in the following pages.

First, we reframed the discussion of the relationships between ENSO/SAM and West Antarctic melt in **probabilistic** terms ("how likely was the melt event to occur given the strong El Niño and the positive SAM") in lieu of a more **deterministic** approach in the original text ("the strong El Niño and the positive SAM caused the melt event"). Opting for a probabilistic approach requires performing a statistical analysis on a sufficiently large number of events, in our case **major** El Niño events. This is clearly not possible if we limit ourselves to the post-1979 period (the period with satellite-based observations of surface melt and reliable atmospheric reanalysis data), which includes no more than three major El Niño events (1982-83, 1997-98, and 2015-16). Incorporating other datasets (e.g., ice cores or climate models) to extend the time series, while desirable, also brings a number of challenges or uncertainties.

Second, among the different options available to us to strengthen our analysis of the relationships between ENSO/SAM and West Antarctic melt, we decided to take advantage of existing output from the Community Atmospheric Model version 4 (CAM4). This output consisted of a set of three 15-year **idealized** CAM4 simulations forced over the ocean with annually-repeating sea surface temperatures (SSTs) characteristic of past strong El Niño events. An additional simulation was forced with annually repeating climatological SSTs to serve as control run. The three El Niño simulations allowed us to investigate the atmospheric response to a total of **45 major El Niño events**, which is substantially more than the three observed since 1979. The results from this additional analysis are discussed in three new subsections ("*Assessing the contribution of EL Niño and SAM*", "*Modeling experiment setup*", and "*Results from climate model simulations*", lines 151–207) at the end of the main Results section of the manuscript. These new results are summarized below.

Based on the CAM4 simulations, we derived a model SAM index. Although the model can output surface melt, some caution is required with this parameter as it is affected by model grid resolution, model temperature biases, and model deficiencies in the placement of key atmospheric features. Given these potential issues, we decided to use surface temperature anomalies as an indicator of melt-prone conditions, rather than surface melt itself (these anomalies were calculated for each simulation by subtracting the long-term monthly means of the control simulation).

We found that the 45 El Niños were associated with 32 **warm events** (71%) and 13 cold events (29%). This finding is consistent with the known positive impact of the El Niño teleconnection pattern in the South Pacific on West Antarctic temperatures. Out of the 32 **warm events**, 15 (47%) occurred during a **negative SAM** phase. Out of the 13 **cold events**, 8 (61.5%) occurred during a **positive SAM** phase. A chi-square statistical analysis shows that the type of event was significantly dependent on the combined states of El Niño and SAM. Furthermore, we also found that, out of the top 10 warmest events, 9

occurred during a negative or neutral SAM phase, while 9 out of the top 10 coldest events occur during a positive or neutral SAM phase.

These model results demonstrate that (i) **strong El Niño events are conducive to conditions favorable to surface melting in the Ross sector of West Antarctica**, and (ii) during strong El Niño events, these conditions are **much more likely to occur when the SAM is in a negative or neutral phase**. Going a step further, given that the SAM was in a predominantly positive phase before and during the January 2016 melt event, we conclude that the state of **the SAM likely mitigated the magnitude of West Antarctic surface melt** during the 2015–16 austral summer. In other words, the 2015–16 melt season would likely have been more prominent had the SAM been in a negative or neutral phase, more commonly associated with El Niño events.

Responses to Reviewer #1

Nicolas et al document an extensive surface melt event during early 2016 in West Antarctica using a combination of remotely sensed melt observations, in situ ground and atmospheric observations from WAIS Divide, and atmospheric and oceanic reanalyses. Using these datasets, the authors attribute the melting to the strong 2015 El Niño and the strongly positive Southern Annular Mode (SAM) during the 2015-2016 austral summer. Based on these inferred links, the authors conclude that extensive WAIS melting may increase in frequency in the future, due to the expected increases in El Niño intensity and positive phases of the SAM.

Understanding the causes and impacts of extensive melting across WAIS is certainly an important endeavor. The authors do a nice job of documenting the melt event via the passive microwave melt time series and its potential direct causes (i.e., warm air intrusion and low level liquid clouds). However, the link between this event, ENSO, and SAM is not robustly presented. To me, this represents a major flaw in the manuscript. For example, there is no statistical analysis between melt and these climate indices necessary to link the phenomena. To establish a link (which would then better support the rather speculative conclusions about future increases in WAIS melt), a statistical analysis is necessary as well as an investigation of the past extensive WAIS melting episodes. Otherwise, we are simply learning about the weather conditions during the 2016 melt event, not the climatic processes and teleconnections ultimately governing or influencing extensive WAIS melting. The fact that the 2016 event was nearly identical to the 2005 event in terms of extent and that there was a lack of a strong El Niño in 2004-2005 suggests that other important factors beyond El Niño are at play.

We agree with the Reviewer that a more robust statistical analysis was desirable to clearly justify our claim of a causal link between the state of ENSO and SAM and the January 2016 melt event. On the other hand, we are also cognizant that the post-1979 period (for which satellite-based melt observations are available) precludes such analysis. While the time series may be long enough (38 years), the period contains only three major El Niño events. These are indeed the type of events most relevant to our analysis (not the full continuum of ENSO states described by the SOI index). Further response to the Reviewer's comments is provided in our Response common to all Reviewers.

I would also suggest toning down statements about how increases in WAIS melt may lead to its future destabilization (e.g., the summary paragraph sentence stating "... the potentially important role that surface melt could play in the disintegration of the WAIS in the coming decades"). While WAIS surface melt events like that in 2016 are anomalous in the satellite record and quite extensive, they are certainly weak relative to melt conditions over Greenland or on the Antarctic Peninsula where surface melt has been linked to hydrofracture.

The sentence mentioning the disintegration of the WAIS was removed from the summary paragraph.

Specific Comments:

Figure 1: Please label each AWS on the map.

Abbreviated AWS labels have been added to Fig. 1a.

Line 59-60: The westward propagation towards the TAM at the onset of "Phase 2" is not very clear in the supplementary figure 1. Perhaps you could indicate the extent of the Ross Ice Shelf to make it more clear where the TAM are in the figure?

We have outlined the contours of the Ross Ice Shelf in all the panels of Supplementary Fig. 1.

Line 65-66: The author state that a positive SST anomaly around 50S/120W contributed to warm air flowing south, citing figure 2b. I have a few questions here (the first two apply to Figure 2c as well):

- **What time period were these anomalies derived from? This needs to be stated.**

These SST anomalies are directly provided by NOAA and are with respect to the 1971–2000 period. The Z500 anomalies are with respect to the 1979–2016 January mean. These additional details have been added to the caption of the figure (formerly Fig. 2, now Fig. 5).

- **Are these anomalies significant? Can you shade areas that are significant?**

The role of the warm SST anomalies is secondary to the role played by the atmospheric circulation. We do not consider the statistical significance of these anomalies as an important piece of information. Adding such information would also overload the figure. Therefore, we decided against showing the statistical significance in Fig. 2b.

- **The link to the positive SST anomaly is not well supported by the presented analysis, in particular given that a larger and negative SST anomaly exists closer to WAIS and over which this "warm" air would have needed to travel. Would these not cancel out the much smaller and isolated area of warm water?**

The Reviewer's comment that "*a larger and negative SST anomaly exists closer to WAIS and over which this "warm" air would have needed to travel*" does not seem to be consistent with the patterns of SST anomalies shown in Fig. 2c. The positive geopotential height anomalies centered near 60°S, 90°W promote **anticlockwise** atmospheric circulation anomalies. As a result, the air does not have to travel over any area with large negative SST anomalies. The Reviewer's comment would be correct if the positive geopotential height anomalies were promoting **clockwise** atmospheric circulation anomalies. To avoid any confusion, we have added a parenthetical note to the text saying that "*the positive geopotential height anomalies near 60°S, 90°W favor anticlockwise motion*" (line 121).

Line 67: How confident are you in the ERA-Interim rainfall data in supplementary figure 2? The rain over WAIS and particularly the rain at 120W and >80S seems very suspect to me. Could this rainfall be corroborated by passive microwave brightness temperatures?

First, we have added a note of caution about the reanalysis data: "*The reanalysis data should be treated with caution*" (line 124). On the other hand, at least two lines of evidence suggest that it did rain over portions of the Ross sector at the beginning of the melt event (lines 124–126). First, **rain was witnessed on by the field party led by Dr. Huw Horgan** (Victoria University of Wellington, personal communication) on the Ross Ice Shelf. Second, **drizzle was detected at WAIS Divide** by a Parsivel optical disdrometer (PAR), which was part of the suite of instruments deployed by the AWARE Project. A figure showing the PAR observations for 11 January 2011 was added to the Supplementary Material (Supplementary Figure 6).

Line 68: Could the blocking event that set up prior to phase 1 have also preconditioned the snow surface for melting (e.g., by increasing shortwave radiation under clear skies)? The atmospheric blocking prior to the warm air advection seems like it might be an important piece of the puzzle.

Near-surface temperature observations from the West Antarctic AWSs prior to January 10, 2016 (onset of the melt event) shown in Fig. 1c suggest that this scenario is unlikely. Indeed, these temperatures remained well below freezing at most sites up until January 10. This is not inconsistent with the presence of the blocking prior to January 10 (visible in Fig. 5b). Indeed, this

blocking was located too far offshore to significantly affect weather conditions in West Antarctica (the latitudinal averaging applied to the geopotential height anomalies does not allow the figure to convey the southward motion of the blocking).

Lines 83-85: It is stated that El Niño-linked blocking and strong westerlies (due to strongly positive SAM) "caused the strong southward warm air advection that contributed to the melt event." This statement of a causal link is not supported by any evidence. Rather, only the occurrence of +SAM and El Niño are established. Furthermore, the meanders in the jet stream around Antarctica (as shown in Fig 2a) are common over short time scales, irrespective of the state of SAM. Thus, in my opinion, the role of SAM (and ENSO) here remains questionable.

As explained in our Response common to all Reviewers, we now discuss the causal relationship between El Niño/SAM and the melt event in probabilistic rather than deterministic terms. Furthermore, the additional work carried out to revise our manuscript led us to reassess the role of the SAM during the melt event. While we initially argued that the positive SAM phase had contributed to the melt event, our analysis suggests instead that the positive SAM phase **mitigated** the magnitude of the melt event. In other words, the melt event likely occurred **despite** rather than **because of** the positive SAM. Finally, regarding the "questionable" role of ENSO mentioned by the Reviewer, we want to underscore (as we do in the manuscript) that the atmospheric circulation anomalies observed over the South Pacific in January 2016 were consistent with those **typically** associated with El Niño events, as documented by a large body of literature.

Lines 89-91: Please expand upon how the clouds affect the radiative fluxes for the reader that might not be familiar. What are the most relevant terms here to ice sheet surface melt - shortwave or longwave fluxes, both? Also, please be more explicit in the physical processes influencing melt that relate to low level liquid clouds.

We changed the text to: *"Clouds exert an important influence on the SEB by modulating the radiative fluxes, **primarily by enhancing downwelling longwave radiation and attenuating incoming solar radiation**" (line 78).*

Lines 100-103 / Figure 3d,e: The plots in figure 3 d,e from 12 January (two days after the start of phase 1) show periods of high backscatter and low depolarization ratio as stated in the text, but they also show the opposite as well. The wording here should be changed accordingly (perhaps state "periods" or "intervals" of high attenuated backscatter...). Perhaps more importantly, the data also suggest non-mixed phase clouds almost as frequently during this interval. Like my comment above regarding Lines 65-66, this seems to be a selective interpretation of the data.

The text was changed to: *"Micropulse lidar measurements (Fig. 3d,e) yielded **periods of high attenuated backscatter (>10 dB) and low depolarization ratios (<10%) below 1 km**" (line 97). Furthermore, the original text only focused on episodes with $10 < \text{CLWP} < 40 \text{ g m}^{-2}$. We have added two sentences to discuss cases where $\text{CLWP} > 40 \text{ g m}^{-2}$: *"We also notice a significant frequency of $\text{CLWP} > 40 \text{ g m}^{-2}$ (Fig. 3c), under which shortwave flux is attenuated and longwave flux is similar to blackbody radiation at the cloud effective temperature. These optically thicker clouds represent a contrast to the Greenland cloud radiative enhancement effect in that they signify a more prominent role of thermal blanketing as a consequence of the warm air advection (Fig. 3a,b)" (lines 106–110).**

Lines 108-109 / Supplementary Figure 3: The ERA-Interim data seem to match up nicely with the field observations from WAIS Divide, however, they also show that much of the ice sheet that underwent melting did not experience similar conditions to those at WD. This raises the question of whether the

observations at WAIS Divide of liquid clouds are relevant to interpreting the cause of the widespread melt event.

In hindsight, we believe that this figure (showing the occurrence frequency of CLWP within 10–40 g m⁻²) was difficult to interpret without any other supporting figures. As a result, three new figures (Fig. 2, Supp. Fig. 2, and Supp. Fig. 3) were added to the manuscript to highlight how the meteorological conditions at WAIS Divide (captured by AWARE observations) were indicative of conditions experienced by a broad portion of the Ross sector of West Antarctica. These new figures display the following information:

- **Figure 2:** Map of ERA-Interim CLWP for 06:00 UTC 12 January 2016 along with two profiles of cloud particle phase from CALIPSO Lidar retrievals. Both ERA-Interim and CALIPSO show widespread occurrence of liquid water clouds over West Antarctica at the beginning of the melt event.
- **Supplementary Figure 2:** Series of daily maps showing the daily mean CLWP estimated by ERA-Interim during during January 9–28, 2016.
- **Supplementary Figure 3:** Same as Supplementary Fig. 2, but with CLWP values binned into three categories (< 10 g m⁻², 10 g m⁻² < x < 40 g m⁻², > 40 g m⁻²) to highlight areas where the radiative enhancement mechanism described by Bennartz et al. (2013) for Greenland (middle category in our scale) may have played a role.

Line 111: Remove "net" as you are referring only to the downward longwave radiation.

Done.

Line 112: The additional energy input has warmed the surface snow temperature, which is reflected in the satellite brightness temperatures (not the energy input directly as it is stated currently). This raises another point: Did you model the SEB at WAIS Divide to calculate snowpack temperature? Did surface melt occur at WAIS Divide? Based on your excess energy (~>30 W m⁻²), it seems that the surface would have been very close to melting (depending on the magnitude of heat loss to the subsurface).

No, we did not model the SEB at WAIS Divide to calculate snowpack temperature. According to SSMIS brightness temperature (T_b) observations, surface melting did not occur at WAIS Divide (see daily T_b anomalies in Supplementary Fig. 1). However, the two T_b observations available each day were 12 hours apart and thus may have missed a brief episode of surface melting. Furthermore, AWS temperature observations (shown in Fig. 1c) rose to 2.1°C on January 11, 2016. Those among the coauthors who were present at WAIS Divide on January 10-12 did experience near-melting conditions. For example, Mr. Jonathan Wille reported that “any non-white surface conducted enough heat to melt the surrounding snow despite the persistent cloud cover.”

Lines 112-114: This sentence is a bit unclear. What is the "positive perturbation" referring to? From looking at Figure 4e, it seems that there remains both a positive perturbation in both the T_b time series and the net SEB during and after phase 1. Also, I'd be skeptical about having a heat flux from deeper layers up to the surface at WD - do you have subsurface temperature profiling to support this claim?

The “positive perturbation” referred to the higher brightness temperatures that persisted from January 10 to January 17 (unfortunately, we don’t have any subsurface temperature measurements from WAIS Divide). We agree with the Reviewer that the switch to larger surface energy gains remains apparent in our SEB measurements as well. Therefore, we decided to

remove the following sentence in the revised manuscript: *“the positive perturbation persists longer than in our calculated energy budget, indicating the heat contribution from the subsurface snow layers.”*

Line 123: "changes in other aspects of the phenomenon" is unclear. Can you be more specific?

We changed the text to: *“Future changes in the intensity of ENSO events are currently estimated with greater confidence than ENSO-related long-distance climate changes, owing in large part to model difficulties in simulating the present-day mean state climate”* (line 217–218).

Line 144: Change "passive microwave emission" to "microwave brightness temperature".

Done.

Line 157: Tedesco 2009 does not go into detail on the method as cited here. Did you generate your own regression coefficients, or use those of Jezek et al (1991) and Abdalati et al (1995) as employed by Liu et al (2006)? If not, the methodology used in this study should be better explained and regression coefficients listed in supplementary information - thus allowing reproduction of the dataset.

We generated our own regression coefficients only for the calibration between SSMI/F-13 and SSMIS/F-17 since, to our knowledge, these coefficients have not been published. To derive these new coefficients, we used brightness temperatures measured over the Ross sector of West Antarctica (land point only) from 1 May through 30 September 2007 (part of the overlap between the F-13 and F-17 missions). The May–September months are chosen for consistency with the cross-calibration between F-11 and F-13 described in Stroeve et al. (1998).

For cross-calibration between older sensors, we used the regression coefficients calculated by Jezek et al. (1991), Abdalati et al. (1995), and Stroeve et al. (1998). All brightness temperatures were adjusted to SSMIS/F-17. To avoid any confusions and ensure perfect reproducibility of our dataset, we have added two tables to the Supplementary Material in which we list:

- the coefficients as they were published or (for F-13 to F-17) as we calculated them ourselves (**Supplementary Table 2**).
- the coefficients that we ended up using to adjust all brightness temperatures to SSMIS/F-17 (**Supplementary Table 3**), which are simply linear combinations of the coefficients listed in Supplementary Table 2.

In addition, the text of the Supplementary Material was changed as follows (lines 251–255): *“To ensure consistency between the different sensors, we adjusted all SMMR and SSM/I Tb data to SSMIS F-17 using the regression coefficients derived by refs. 40–42. The only exception was for the adjustment between SSM/I F-13 and SSMIS F-17, for which we derived our own coefficients (see Supplementary Fig. 8). All coefficients used in our adjustment procedure (along with their references) are listed in Supplementary Tables 2 and 3.”*

Reviewer #2

The authors present an interesting case study of widespread surface melt over Western Antarctica and the Ross Ice Shelf. The authors combine detailed surface-based observations from different Automatic Weather Stations, the AWARE Radiation Experiment site, satellite observations, and reanalysis data to characterize the melt event. The manuscript is well written and easy to follow. The characterization of the synoptic situation that led to the surface melt is in my view the strongest part of the manuscript. Advection of warm air was driven by a strong and persistent high-pressure ridge slightly west of the investigation area. This ridge persisted over several days and allowed for warm air to be advected southwards over the investigation area.

Using the combination of AWARE surface observations and ERA reanalysis the authors also find thin, liquid-bearing clouds to enhance surface melt. The detailed radiation measurements presented in Figure 3 substantiate this point for AWARE for the period 10-13 January but do not provide data for the period after January 19. It would have been interesting to see the measurements also for 'Phase 2' of the melt event (even though during that later time period the event had moved to the east).

Unfortunately, the AWARE campaign ended on January 19, therefore no AWARE observations are available beyond that date.

In addition, ERA reanalysis (Figures S1 and S2) shows the occurrence of liquid precipitation and widespread liquid clouds over the investigation area. While the findings based on ERA reanalysis certainly are in agreement with the authors claim, it is unclear to me how much one can trust ERA over the southern high latitudes. NWP models are known to underestimate the amount of low-level liquid in southern clouds, in particular in regions of subsidence (Naud et al., 2014). Mid-representation of clouds is also known to lead to large biases in the radiation balance (e.g. Trenberth and Fasullo 2010). Such biases in model-derived cloud parameters could result in either an under- or overestimation of the role of liquid clouds in the context of this study. In order to substantiate the point made by the authors, it would be worthwhile to try augmenting the study with additional satellite data, preferably with MODIS observations of liquid clouds and/or space-borne lidar observations from NASA's CALIOP mission.

We followed the Reviewer's advice and added a new figure (Fig. 2) to the main text showing two profiles of cloud particle phase from CALIPSO/CALIOP products alongside a map of ERA-Interim CLWP estimates for 12 January 2017. These two datasets show widespread occurrence of liquid water clouds over West Antarctica at the beginning of the melt event and thus support AWARE observations at WAIS Divide. A discussion of this new figure is provided at lines 81–94.

The largest issue I have with the manuscript lies in the implied relation between El Niño and the observed warming. I have to admit that I am not an expert in large-scale teleconnections, so my following comments should be taken with a grain of salt. In my view, the manuscript does not necessarily justify the authors' claim that the warm air advection arose "from the conjunction of a record El Niño event and strong circumpolar westerly winds". Clearly, both a record El Niño and strong westerly winds existed. However, the relation between El Niño and the warming appears to be not actually found in this study. Rather, the study relies on earlier studies that appear to have shown the effect. See lines 71–73 in the manuscript and references 12 and 18 therein. New mechanisms or teleconnections relating El Niño to melting in Antarctica are in my understanding not discussed in the study.

We agree that the role of "strong circumpolar westerly winds" (i.e., positive SAM) was not properly substantiated in our initial submission. As we explain in our Response common to all

Reviewers, the additional work conducted as part of the revisions to our manuscript led us to reassess the contribution of the SAM to the melt event. Indeed, we argue, based on our new model results, that the positive SAM likely **mitigated** the magnitude of West Antarctic surface melt during the 2015–16 austral summer.

Furthermore, we do not entirely agree with the Reviewer’s comment about “the relation between El Niño and the warming [not being] actually found in this study.” As we underscore in the manuscript, the presence of the positive geopotential height anomalies over the South Pacific sector of the Southern Ocean in January 2016 was a **typical** signature of El Niño events. This statement is supported by a significant body of literature, based both on observations and model simulations. Granted, the precise characteristics (location, magnitude, timing) of this El Niño teleconnection has varied between the various El Niño events that have taken place since 1979. But the presence of these typically atmospheric circulation anomalies in January 2016 was a **strong indication** that the concurrent strong El Niño did indeed play a role in the West Antarctic melt event.

Another confusing issue arises from the lack of consistency between the observed melt events Jan-83, Jan-92, Jan-05, and Jan-16 and the corresponding SAM and ONI anomalies. From Figure 2 we have:

- **Jan-83: SAM negative, ONI positive**
- **Jan-92: SAM neutral, ONI positive**
- **Jan-05: SAM neutral, ONI slightly positive/neutral**
- **Jan-16: SAM positive, ONI positive**

There are also years (e.g. Jan-10) where ONI is slightly positive/neutral and no melt occurs. I think this generally substantiates the author’s own finding that “Further research is needed to better understand [...] mechanisms behind major West Antarctic melt events [...]”.

We agree with the Reviewer that the lack of consistency between West Antarctic melt, ENSO, and SAM was a source of confusion. We have addressed this concern in the following manner:

- We now underscore that the relationship between the three phenomena is complex and that it is not possible to establish with certainty whether the January 2016 was caused by the strong El Niño or the strong SAM. Instead, following a **probabilistic** approach, we attempt to determine how likely the melt event was to occur given the ENSO and SAM conditions in January 2016.
- We highlight visually the relationships between West Antarctic melt, ENSO, and SAM with a new figure (Fig. 7). It consists of a bar chart showing the values of the West Antarctic melt indices, SAM Index, and SOI for each austral summer since 1979. The originality of the plot is to have the years (on the x-axis) ranked-ordered based on the melt indices. However, the figure also highlights the overall directions of the relationship as we described them in our initial submission; that is, a general tendency for
 - o less melt to occur during La Niña-like conditions ($SOI > 0$) and a positive SAM phase.
 - o more melt to occur during El Niño-like conditions ($SOI < 0$) and a negative SAM phase.

This figure confirms that these relationships are complex, do not lend themselves easily to prediction, and as such are better suited for a probabilistic approach.

Finally, if my above assessment holds, it would also call in question the speculative last sentence of the introduction where the authors state that surface melt could potentially (sic!) play an important

role in the disintegration of the WAIS in the coming decades. It is of course true that “potentially” surface melt can play an important role. However, in my view, the study presented here does not necessarily provide strong evidence one way or another.

The sentence “This underscores the potentially important role that surface melt could play in the disintegration of the WAIS in the coming decades” was removed from the abstract.

Reviewer #3

This paper describes an extensive melt event across West Antarctica that occurred in January 2016. The event occurred during a strong blocking episode that resulted in the advection of warm, maritime air into West Antarctica and the formation of mixed-phased clouds. The authors argue that the combination of relatively warm air and enhanced downward longwave radiation from the cloud cover contributed to the duration and intensity of the surface melting. Further, the blocking episode is described as part of a teleconnection pattern driven by a strong El Niño event in the tropical Pacific. The most unique aspect of this paper is the inclusion of field measurements of surface energy balance and clouds taken during the AWARE campaign. Such measurements are rare and their use in conjunction with satellite data and atmospheric reanalysis makes this paper particularly noteworthy. The paper is very well-written, easy to read, and concise.

The weakest link is the climate change connection - is this one event a sign of things to come? Will surface melting be a significant contributor to ice sheet mass loss? Has this type of melt event occurred in the past? The answers to these questions are difficult but I think are what would make it a strong Nature Communications paper as opposed to a noteworthy field study.

If I have one major recommendation to improve the paper, it is to flesh out one of these questions a bit more - perhaps include either some historical context from ice cores or some results from climate models. Since El Niño events are intermittent and it is hard to predict whether a given event will produce the right teleconnection pattern, it is hard to say whether the link of El Niños to melting episodes is significant.

The question we decide to flesh out was not explicitly listed by the Reviewer. It is about shedding additional light on the linkages between ENSO/SAM and West Antarctic melt. We believe that gaining a solid understanding of these linkages (to which we hope our manuscript can contribute) is part of the prerequisites for investigating future changes in Antarctic surface temperatures and Antarctic surface melt events. Our revised manuscript includes several new subsections where we expand our discussion of the contribution of El Niño/SAM to the January 2016 melt event, building on our analysis of idealized climate model simulations (see details in our Response common to all Reviewers).

I am wondering if just the slow atmospheric warming associated with climate change would be sufficient (by say 2100) to significantly increase the probability of summer melt events, regardless of what happens to the frequency or intensity of El Niños? Model results may not be reliable enough to make the link to surface melting directly, but they may be able to shed light on projected warming trends and when the forced climate change signal in surface air temperature will emerge above natural variability, and when this signal plus a strong El Niño would surely produce near-freezing temperatures on WAIS.

We believe that we are not yet able to answer the question (whether “*the slow atmospheric warming associated with climate change would be sufficient to significantly increase the probability of summer melt events, regardless of what happens to the frequency or intensity of El Niños*”). Using existing future climate model projections (particularly from the CMIP5 archive) to investigate future changes in Antarctic surface temperatures and surface melt comes with a number of caveats. Large uncertainties exist regarding future warming trends in and around Antarctica, owing for example to significant uncertainties in the future evolution of Antarctic sea ice. As a result, this question is one that we decided not to address in our manuscript in order to

maintain our focus on the relationships between large-scale mode of climate variability and West Antarctic surface melt.

Minor comments

Lines 67-68 and Supp figure 2: Is there any evidence that it actually rained on the WAIS during this event, or is it just that the reanalysis model forecasted rain? Were there any ground observers at WAIS Divide to corroborate the forecasted rain? In any event, the rain does not look very extensive compared to the area of melting. I'm not really sure that these data and the argument about preconditioning of the snow add much to the story.

Yes, there is evidence that it rained in various parts of the Ross sector at the beginning of the melt event (see lines 124–126). First, one **field party** present on the Ross Ice Shelf at that time did witness rain. This piece of information was added to the text as a personal communication. Second, **drizzle was detected at WAIS Divide** by a Parsivel optical disdrometer (PAR), which was part of the suite of instruments deployed by the AWARE Project. A figure showing the PAR observations for 11 January 2011 was added to the Supplementary Material (Supplementary Figure 6).

Lines 133-142: I think more explanation is needed here. Why did the two studies reach such different conclusions? Did they use different climate change scenarios (RCP 4.5 vs RCP 8.5), different algorithms to calculate melt, or different ice sheet or climate models? Does the present study shed any light on this issue?

The main explanation for the contrasting results between the two studies is that they used different models and different modeling strategies to reach their conclusions. They both considered the RCP8.5 scenario, therefore the choice of future climate change scenario is not at issue here.

- **Trusel et al. (2015)** used the RACMO2 regional climate model driven (successively) by five global climate model simulations from the CMIP5 archive. Then, they **inferred** the future state of the ice shelves based on the evolution of the 2-meter temperature and meltwater production at the surface.
- **DeConto and Pollard (2016)** used the RegCM3 regional climate model nested inside the GENESIS v3 global climate model, along with ocean temperatures from the CCSM4 global climate model, to drive their ice sheet/ice shelf model. As such, they directly **simulated** the future behavior of the ice sheet and ice shelves.

We believe that we are not in a position to pinpoint the precise reasons for why the two studies came to such different conclusions. These differences are likely due to a **combination of multiple factors**. Furthermore, our study of the January 2016 melt event does not allow us to lend more credence to either one. Finally, it is worth pointing out that neither of them discuss the role of large-scale modes of climate variability such as ENSO and the SAM and their role in future Antarctic temperature changes.

REVIEWERS' COMMENTS:

Reviewer #1 (Remarks to the Author):

I am pleased that the authors have carefully considered and responded to all of my original comments/suggestions. The authors have implemented numerous changes and included important new analyses, text, and figures that better support their conclusions. In particular, I appreciate the new idealized model simulations the authors have performed and presented in the revised manuscript. These simulations add important statistical context for strong El Nino and SAM events and the effects of these on near surface air temperatures and melt in West Antarctica.

This is now a well-supported and well-written paper suitable for publication in Nature Communications. I look forward to seeing it in print in the near future.

Reviewer #2 (Remarks to the Author):

The authors have added substantial analysis and have fully addressed my initial concerns about their hypothesized relation between ENSO and surface melt. The addition and careful analysis of a series of SST-forced climate model runs proves very valuable and allows the authors to draw firm and statistically significant conclusions about their postulated link.

The paper is highly relevant in the context of Antarctic ice loss.

It can be published as is.

Reviewer #3 (Remarks to the Author):

2nd Review of "January 2016 extensive summer melt in West Antarctica favored by strong El Nino" by Julien P Nicolas et al.

I think the authors have done a satisfactory job of addressing my and the other reviewers' concerns about the original manuscript. They have toned down the claim about the 2016 melt event being a precursor of things to come on the WAIS. They have also addressed the ENSO-SAM-melt relationships more thoughtfully and thoroughly. As a result of these revisions, the paper is less sensational than it was before, although the analysis is more sound, and I think the paper strikes the appropriate tone.